# Investigating Persistent Sympathovagal Dysregulation Following a Complex Dual Task in Concussed Athletes

**DOI:** 10.3390/jfmk10020115

**Published:** 2025-03-29

**Authors:** Mathieu Bolduc, Gabriel Lavoie, Veronik Sicard, Julien Lépine, Dave Ellemberg

**Affiliations:** 1Department of Psychology, University of Montreal, Montréal, QC H3T 1J4, Canada; mathieu.bolduc.2@umontreal.ca (M.B.);; 2CHEO Research Institute, Ottawa, ON K1H 5B2, Canada; veronik.sicard@gmail.com; 3Department of Family and Emergency Medicine, University of Montreal, Montréal, QC H3T 1J4, Canada; 4Department of Kinesiology, University of Montreal, Montréal, QC H3T 1J4, Canada

**Keywords:** concussion, heart rate variability, dual task, cognition, autonomic nervous system

## Abstract

Background/Objectives: Dual tasks are increasingly being employed in research on concussion, since they provide a somewhat more realistic representation of the demands athletes face on the field. While single cognitive and motor tasks have revealed persisting autonomic alterations in concussed athletes, the unique autonomic response required by a dual task remains unexplored in this population. The purpose of this study was to evaluate autonomic responses in asymptomatic athletes with a history of concussion (m = 5.46 months ± 2.00) following a complex dual task. Methods: Heart rate variability (HRV), a biomarker of autonomic regulation, was measured in 34 athletes (16 concussed, 18 controls) aged 17 to 24. HRV data were collected using the Polar H10 chest belt. Five-minute segments were extracted under four conditions: rest, following a cognitive task (switch task), after a dual task combining both motor and cognitive components, and after the same dual task preceded by 20 min of aerobic exercise. A series of 4 × 2 mixed-design ANOVAs were conducted to assess the differences between the conditions and groups. Results: The results indicated a significant increase in a global marker of HRV (i.e., the standard deviation of normal-to-normal intervals (SDNN)) following the switch task compared to rest (*p* = 0.014) only in concussed participants. These results suggest that the switch task may stimulate frontal regions and promote a parasympathetic response, as reflected by the rise in HRV. Notably, the effect of the switch task disappeared when combined with the motor component of the dual task, whether it was preceded by aerobic exercise or not. Conclusions: The dual task results indicate potential competing mechanisms between the motor and cognitive components of the task, which future studies using similar protocols should consider. Meanwhile, the switch task appears sufficiently demanding to reveal autonomic alterations, which, when measured through HRV, may constitute a relevant clinical tool for assessing athletes’ readiness to return to sport and academic study.

## 1. Introduction

Sport-related concussions (SRCs) have long been recognized as a significant public health concern and have garnered increased attention in recent decades. Predominantly occurring in contact sports, SRCs are common injuries among athletes [1,2,3,4]. An epidemiological study encompassing 23 NCAA sports found that 3497 SRCs were reported from 2014 to 2019 [5]. Meanwhile, other sports organizations also reported notable concussion rates per season, with 177 concussions documented in the NFL during the 2018–2019 season, 68 in World Rugby’s 2019 season, 67 in FIFA’s 2014 season, and 46 in the NHL’s 2017 season [6,7,8,9]. However, the actual incidence of concussions may be even higher, as under-reporting is a well-documented issue among athletes [10,11,12]. For instance, Kerr et al. [12] found that 33% of former collegiate athletes admitted not disclosing at least one concussion during their career, primarily due to fear of being removed from play or believing the injury was not serious enough. While some of the symptoms experienced during the acute phase of concussion may appear benign to some athletes, the persistent consequences that may unfold largely justify the temporary removal of athletes from competition [13].

The literature on the long-term consequences of concussions indicates persisting alterations in physical, emotional, and cognitive functions, alongside an increased risk of neurodegenerative diseases [14,15]. Specifically regarding cognitive impairments, a meta-analysis indicates that concussed athletes may experience significant deficits in executive function (EF), including strategy generation/regulation, verbal set-shifting, and interference management, as well as impairments in prospective working memory and response inhibition [16]. For instance, a study by Sicard et al. [17] found that 21% of athletes who suffered a concussion on average 29 months prior to testing exhibited significant deficits in higher-order EF processes during a cognitive task (known as the switch task) [18]. Notably, those alterations were only present when athletes performed an aerobic exertion task prior to the cognitive task. The inclusion of exertion tasks in concussion assessments is increasingly recommended, as they can reveal deficits that remain undetectable at rest [17,19,20,21,22]. While the cumulative evidence of persistent alterations following a concussion is concerning, evidence indicates that the autonomic nervous system (ANS) may also undergo lasting alterations following a concussion [21,23,24].

The primary function of the ANS is to regulate the activity of most organs and to maintain homeostasis by coordinating activities across various systems [25]. It is divided into two main subsystems: the parasympathetic nervous system (PSNS), which promotes relaxation and energy conservation, and the sympathetic nervous system (SNS), which prepares the body for action [26]. For athletes, the ANS is vital in regulating physiological responses to physical exertion and stress, making any alterations particularly worrisome for performance and recovery [27].

Anatomically, the ANS involves both cortical and subcortical regions. Some of those regions are key components to the central autonomic network (CAN), which integrates autonomic functions with cognitive and emotional processes [28,29]. The prefrontal cortex also indirectly influences autonomic regulation by modulating activity within the CAN through extensive connections with limbic and subcortical structures [30]. Notably, these regions are commonly affected by concussions [31,32,33,34,35,36], which has led researchers to investigate their impact on the ANS [24,37,38]. To assess autonomic activity, researchers have commonly utilized heart rate variability (HRV), a widely accepted non-invasive biomarker that reflects fluctuations in the sympathovagal dynamic [39].

HRV refers to the variation in the time intervals between consecutive heartbeats, which can be identified through an electrocardiogram (ECG) [39,40]. Higher HRV usually signifies greater parasympathetic (vagal) activity and indicates a healthy, responsive ANS, while lower HRV typically reflects sympathetic dominance and reduced autonomic flexibility, which may be associated with stress, fatigue, or impaired regulatory mechanisms [39].

Studies assessing HRV in recently concussed athletes found altered autonomic responses at rest and following physical or cognitive exertion [37,38,41,42,43,44,45,46,47]. For example, Gall et al. [37] reported that concussed athletes exhibited significantly lower HRV during an aerobic exercise protocol compared to healthy peers, highlighting the impact of concussion on autonomic regulation during physical stress.

Relatively few studies examined HRV in the chronic phase of concussion [21,23,24,45,46,48]. Abaji et al. [24] assessed HRV in athletes approximately 20 months post-concussion. They found that concussed athletes exhibited significantly lower HRV during a handgrip anaerobic task compared to their non-concussed peers, indicating persistent autonomic dysfunction. Similarly, Memmini et al. [21] reported that concussed athletes who suffered a concussion 25 months prior took longer to return to baseline HRV following aerobic exercise than their peers without a concussion history. These results suggest that autonomic dysfunctions may persist long after the initial injury, particularly when the ANS is challenged by physical stress.

Some literature suggests that increased cognitive load in healthy individuals may result in a shift toward sympathetic dominance and decreased HRV [49,50,51]. Hence, since cognitive load appears to challenge the ANS by stimulating its sympathetic response, it may serve as a useful method for investigating autonomic function in concussed athletes. To obtain a better assessment of the autonomic challenges athletes are faced with on the field, tasks involving cognitive processes relevant to sport performance should be prioritized [52,53,54]. One such task is the switch task, which engages higher-order EF processes, such as sustained attention, working memory, inhibitory control, and cognitive flexibility [18,55,56,57]. In one study, Harrison et al. [23] used a switch task to compare the autonomic responses of chronically concussed athletes (injured 24 months prior) with those of athletes without a history of head injury. During the task, concussed athletes exhibited higher HRV values compared to their healthy peers. This uncharacteristic increase in HRV following a cognitive task was also reported in another study using a different cognitive task on athletes in the acute phase of concussion [44]. Notably, those results contrast with the results of studies using a physical task where a decrease in HRV value was observed for concussed athletes [21,24,37,38].

It should also be mentioned that, in the chronic phase of concussion, there are generally no significant alterations in HRV at rest [21,23,24,45]. This contrasts with the acute phase, where HRV often shows marked changes, even in the absence of physical or cognitive exertion [43,44,45]. This indicates that assessing HRV under challenging conditions, such as physical or cognitive stressors, is important for understanding the long-term effects and recovery process in individuals during the chronic phase of the injury.

Although both cognitive and physical exertion seem to adequately reveal autonomic dysfunctions, real-life situations often demand the simultaneous management of cognitive and physical challenges. For example, athletes may need to make rapid, complex decisions while sprinting or maintaining physical endurance during play. These scenarios place unique and multifaceted demands on the autonomic nervous system. To provide a more accurate representation of the autonomic demands athletes face, such conditions should be evaluated concurrently [58].

Scientific paradigms combining both cognitive and physical elements are commonly known as dual tasks and have increasingly been employed in research on concussion. Such tasks have shown to effectively uncover subtle and persisting alterations in stability, gait, and precision in concussed athletes that may not be apparent during single-task assessments [59,60,61]. For instance, Howell et al. [60] demonstrated that concussed athletes exhibited gait and balance control impairments during dual-task assessments, which could persist months after the injury. Further supporting these findings, a recent functional near-infrared spectroscopy (fNIRS) study revealed altered neural recruitment patterns during dual tasks in concussed athletes [62]. Studying HRV under a dual-task paradigm may, therefore, provide a unique opportunity to evaluate the autonomic response of concussed athletes, as their body manages both cognitive and physical demands, which ultimately raises the question of how a concussion may leave persisting alterations on the ability of the ANS to operate under a complex task.

Given the unique nature of the autonomic demands athletes face during competition and the limited amount of literature exploring these dynamics, it is essential to investigate the long-term effects of concussion on the ANS using paradigms that integrate both cognitive and physical demands. The present study aims to address this gap by examining the relationship between SRC and cardio autonomic regulation by comparing HRV values of student athletes with and without a history of concussion following a dual-task protocol that combines a complex cognitive task with a continuous motor task. We further explored whether a 20 min aerobic exertion condition prior to the dual task might influence cardio autonomic regulation. Based on previous studies presenting contrasting effects of cognitive and physical tasks, we hypothesize that athletes with a history of concussion will exhibit reduced parasympathetic activity following the dual task compared to those of the non-concussed group. The aerobic exertion condition should further strain the ANS.

## 2. Materials and Methods

### 2.1. Participants

This study was conducted with the approval of the Clinical Research Ethics Committee of the Université de Montréal, in accordance with the ethical standards established in the Declaration of Helsinki. To obtain a large enough sample, we recruited participants from various sports and institutions. A total of 34 athletes participated in this study, comprising 16 individuals with a history of concussion (injured 0.9 to 8.9 months prior) and 18 control subjects, all aged between 17 and 24 years. We recruited participants from six CEGEPs (general and professional teaching colleges—a level between high school and university that is unique to the Québec education system) and three university sports teams located in Montreal. Although both male and female athletes were initially recruited for this study, due to methodological considerations—discussed further in the limitations section—only data from male athletes were included in the analyses. Demographic details are summarized in Table 1.

Note that the HRV data presented here were collected as part of a larger dataset, where results on dual-task performance and concussion were previously reported, but the HRV data were not [63]. While their study primarily focused on gait and executive functions, our analysis is centered on HRV. Thus, the present study is the first original publication to examine autonomic responses, as measured by HRV, in concussed athletes following a dual-task performance.

Recruitment was facilitated through direct contact with athletic directors, team coaches, and referrals from fellow athletes. The recruitment period spanned from January to May 2023. To be eligible, concussed athletes were required to be asymptomatic and to have fully returned to their sport activities. Only those who sustained an SRC within the past 12 months were included.

Exclusion criteria included having a neurological or neurodevelopmental disorder, prior head trauma other than SRC, a history of brain surgery, taking medication with psychoactive effects, or having an unhealthy body mass index (BMI). Reported use of illicit substances, like stimulants, opiates, marijuana, sedatives, or drinking more than ten alcohol consumption units per week, also disqualified participation in the study. Finally, being color blind also excluded participation considering the nature of the task.

### 2.2. Questionnaires and Instruments

#### 2.2.1. Questionnaires

During the initial visit, participants underwent a structured interview to confirm eligibility and provided written informed consent. To control for potential confounding effect of anxiety or mood on HRV, we administered the following standardized questionnaires:Beck Anxiety Inventory (BAI) (BAI-16; Pearson assessment, Toronto, ON, Canada): assesses the severity of anxiety symptoms experienced over the past week.Beck Depression Inventory-II (BDI-II) (BDI-II-19; Pearson assessment, Toronto, ON, Canada): evaluates the presence and intensity of depressive symptoms over the past two weeks.Wender Utah Rating Scale (WURS) (WURS-48; Salt Lake City, UT, USA): retrospectively screens for childhood symptoms of attention-deficit/hyperactivity disorder (ADHD).

Participants scoring above the predetermined thresholds on any of these assessments were excluded, as elevated anxiety, depression, or ADHD symptoms can influence HRV measurements [64,65,66,67].

We administered a comprehensive general information questionnaire to collect data on variables such as age, education level, sport played, as well as to confirm adherence to inclusion and exclusion criteria.

We asked all participants if they ever had experienced any concussion-like symptoms following a blow to the head, neck, or body. We then presented a list of clinical symptoms from the Sports Concussion Assessment Tool (SCAT-5) [68]. We assigned participants who reported no incidents consistent with a concussion diagnosis to the control group. For those with a suspected concussion history, the Post-Concussion Symptom Scale (PCSS) from the SCAT-5 was utilized to quantify current and past symptoms. We only included asymptomatic athletes with a medical diagnosis of concussion within the past year in the concussion group. Although data from post-concussion symptoms were obtained, we did not account for it in the analysis, as it was retrospective.

#### 2.2.2. Curve Trainer Woodway Treadmill

Participants performed walking and running tasks on a Curve Trainer Woodway Treadmill (Woodway USA, Waukesha, WI, USA), a non-motorized treadmill that allows for self-paced movement. The treadmill’s curved design enables users to control their speed by adjusting their position on the belt.

#### 2.2.3. Switch Task Color-Shape

The version of the switch task used in this study was developed and validated in our lab and has proven effective in highlighting executive function deficits in athletes with a history of concussion [17,22,57]. Participants responded by pressing either the right or left button on a controller attached to the treadmill. The task was structured into three phases.

Homogeneous Color Phase: Participants responded to stimuli based solely on color, disregarding shape. Sixty stimuli were presented to the participants and lasted about 126 s.Homogeneous Shape Phase: Participants responded based on shape, ignoring color. Sixty stimuli were presented to the participants and lasted approximately 126 s.Heterogeneous Phase: Participants alternated between color and shape rules depending on the stimulus outline—solid outlines indicated the color rule, while dotted outlines signaled the shape rule (see Figure 1). The heterogeneous phase contained 120 stimuli and was administered twice, for a total duration of about 5 min.

An alternative version of the task was created by reversing the association between the stimulus outline and the response rule to mitigate practice effects across sessions [69]. We instructed participants to respond as quickly and accurately as possible within a 2000 ms time limit for each stimulus.

#### 2.2.4. Polar H10 Heart Rate Monitor

Heart rate variability data were collected using the Polar H10 heart rate monitor (Polar, Kempele, Finland), which provides high-resolution ECG recordings at a sampling rate of 1000 Hz. The monitor was secured around the participant’s chest and synchronized with a tablet via Bluetooth for data acquisition using the Kubios software application (version 3.4; Biosignal Analysis and Imaging Group, Kuopio, Finland).

### 2.3. Procedure

To mitigate the effects of learning and the stress induced by the initial visit, we counterbalanced the first two sessions. Half of the participants performed the dual task on their first session, followed by the switch task alone in their second session. Conversely, the other half performed the switch task first followed by the dual task on their second visit. Although we encouraged participants to schedule their sessions at similar times, their academic obligations prevented us from fully controlling for potential circadian influences. However, since this was distributed randomly across both groups, its impact is likely minimized.

The first session proceeded as follows: Participants signed consent forms and completed the general information questionnaire, BAI, BDI-II, and WURS. Upon completion, we instructed participants to secure the Polar H10 monitor around their chest, which was verified by the administrator. We then instructed athletes to sit still for 5 min, then to stand for 10 more minutes. To familiarize themselves with the equipment, we invited them to step on the treadmill and to walk at a pace as close to 6.5 km/h as possible for 3 min. Participants then completed one of the two following conditions, depending on the counterbalancing:Dual task condition: We instructed participants to perform the switch task while keeping a 6.5 km/h pace. They finally had to stand still for another 10 min.Switch task condition: Participants performed an alternative version of the switch task but while standing stationary. The session concluded with another 10 min standing.

In the third and final session, participants stood for 10 min, then transitioned to a 20 min jogging at 80 to 90% of their theoretical max heart rate [70]:HRmax = 208 − (0.7 × age)(1)

This targeted HR aligns with the recommendations proposed in the Zurich return to play protocol [71]. Participants then performed the dual task with the original version of the switch task. Finally, we asked them to stand still for 10 more minutes.

For the analysis, we used ECG segments of the last five minutes of the resting standing phase at the beginning of the first session and the first 5 min of rest following the completion of the tasks for every session. We compensated participants with USD 30 for each session.

### 2.4. Outcome Variables and Data Analysis

#### 2.4.1. HRV Data

We processed raw ECG data using Kubios Scientific (version 3.4; Biosignal Analysis and Imaging Group, Kuopio, Finland). We utilized the software’s built-in features, including R-wave correction, QRS detectors, and beat-to-beat analysis, for data treatment. We then normalized the R-R intervals by visually inspecting and removing any ectopic beats or artifacts. The respiration rate was estimated using the Kubios algorithm, which incorporates both ECG and R-R interval information. Time domain variables included the root mean square of successive NN interval differences (RMSSD) and the standard deviation of NN intervals (SDNN), while spectral variables, obtained through a fast Fourier transformation included low-frequency (LF) power, high-frequency (HF) power, LF/HF ratio, and total power (TP). We also assessed sample entropy (SampEn) from the non-linear domain to evaluate the complexity and predictability of ECG segments. We selected these specific metrics to capture a broader spectrum of autonomic regulation indicators. Employing commonly reported parameters also enhances comparability with existing studies.

#### 2.4.2. Statistical Analysis

We conducted the statistical analysis using IBM SPSS Statistics for Windows, version 28.0 (IBM Corp, Armonk, NY, USA). We set Alpha at 0.05 for all analysis. Prior to analysis, we screened data for missing values and outliers. We assessed normality using the Shapiro–Wilk test and by examining skewness and kurtosis. When the assumption of sphericity was violated, we applied the Greenhouse–Geisser correction.

We performed independent sample *t*-tests to compare demographic variables between the concussion and control groups, including age, body mass index (BMI), and other characteristics that might affect the outcomes. We conducted repeated measure analyses of variance (ANOVA) on breathing and heart rate to ensure that the results were not influenced by these factors.

We performed a series of 2 × 4 repeated measures ANOVAs on each outcome variable, with “history of concussion” as the grouping variable. The four conditions included the last 5 min of the standing resting phase (resting phase), the 5 min following the switch task performed alone (switch task phase), the 5 min following the dual task (dual task phase), and the 5 min following the dual task with the exertion condition (dual task with exertion phase). We calculated effect sizes using partial eta squared (η^2^) and Cohen’s d to assess the magnitude of observed effects. In cases where we found significant main effects or interactions, we applied post hoc analyses with Bonferroni correction to adjust for multiple comparisons and to identify specific group or condition differences.

## 3. Results

### 3.1. Demographic Comparisons

We conducted independent samples *t*-tests to compare demographic variables between the concussion and control groups. There were no significant differences in body mass index (BMI) (*p* = 0.948, F[32] = 0.004), height (*p* = 0.171, F[32] = 1.97), or time since playing sport (*p* = 0.884, F[32] = 0.22). However, there was a significant difference in age between the groups (F[32] = 11.16; *p* = 0.002), with the concussion group being approximately 1.5 years older than the control group.

The repeated measure ANOVA for both respiration rate (F[2.63, 0.00] = 0.84, *p* = 0.956) and HR (F[2.40, 97.98] = 0.925, *p* = 0.416) did not reveal significant differences between groups, suggesting that they did not play an influential role on HRV.

### 3.2. HRV Variables

#### 3.2.1. SDNN

The repeated measures ANOVA for SDNN revealed a significant condition x group interaction (F[2.60, 621.71] = 4.00, *p* = 0.014) with a moderate effect size (η^2^ = 0.11). We also found a significant main effect of condition (F[2.60, 9066.33] = 58.26, *p* < 0.001), but there was no main effect of group (F[1, 6.73] = 0.013, *p* = 0.912). Post hoc analyses revealed a significant increase in SDNN for concussed athletes between the resting phase and the switch task phase (*M* = 18.11, *p* < 0.001, Cohen’s *d* = 4.36), indicating a very large effect. No such change was observed in the non-concussed group (*M* = 0.34, *p* = 1.000) (see Figure 2).

We found no condition x group interaction or main effect of group for any of the other HRV variables, as seen below. 

#### 3.2.2. RMSSD

The repeated measures ANOVA for RMSSD did not reveal a significant condition x group interaction (F[2.37, 336.96] = 2.36, *p* = 0.092). We observed a significant main effect of condition (F[2.37, 5851.83] = 40.90, *p* < 0.001), but no main effect of group (F[1, 11.00] = 0.029, *p* = 0.867) was present. This indicates that RMSSD was the same for both groups, and it only changed as a function of condition.

#### 3.2.3. LF Power

The repeated measures ANOVA for LF revealed no significant condition x group interaction (F[2.74, 49.24] = 0.60, *p* = 0.60). We observed a significant main effect of condition (F[2.74, 561.17] = 6.89, *p* < 0.001), but we found no main effect of group (F[1, 203.57] = 0.932, *p* = 0.342). This indicates that LF was the same for both groups, and it only changed as a function of condition.

#### 3.2.4. HF Power

The repeated measures ANOVA for HF revealed no significant condition x group interaction (F[2.70, 27.03] = 0.515, *p* = 0.654). We observed a significant main effect of condition (F[2.70, 204.63] = 3.90, *p* = 0.014), but we found no main effect of group (F[1, 290.40] = 2.185, *p* = 0.149). This indicates that HF was the same for both groups, and that it only changed as a function of condition.

#### 3.2.5. LF/HF Ratio

The repeated measures ANOVA for the LF/HF ratio revealed no significant condition x group interaction (F[2.93, 26.60] = 2.09, *p* = 0.108). We found no significant main effect of condition (F[2.93, 23.28] = 1.83, *p* = 0.149) or main effect of group (F[1, 15.14] = 0.385, *p* = 0.539). This indicates that the LF/HF ratio was the same for both groups, and that it did not change as a function of condition.

#### 3.2.6. TP

The repeated measures ANOVA for TP revealed no significant condition x group interaction (F[2.39, 8,155,360.46] = 2.83, *p* = 0.056). A significant main effect of condition was present (F[2.39, 54,269,828.17] = 18.82, *p* < 0.001), but we found no main effect of group (F[1, 491,672.63] = 0.08, *p* = 0.779). This indicates that TP was the same for both groups, and that it changed only as a function of condition.

#### 3.2.7. SampEn

The repeated measures ANOVA for SampEn revealed no significant condition x group interaction (F[2.24, 0.04] = 0.571, *p* = 0.587). We observed a significant main effect of condition (F[2.24, 0.90] = 12.77, *p* < 0.001), but we found no main effect of group (F[1, 0.007] = 0.25, *p* = 0.876). This indicates that SampEn was the same for both groups, and it changed only as a function of condition.

## 4. Discussion

The purpose of the present study was to compare HRV between chronically concussed athletes and those without a history of concussion under the following conditions: at rest, after completing a switch task, following a dual-task protocol combining motor and cognitive demands, and after the same dual-task protocol preceded by a 20 min aerobic exercise session. Concussed athletes exhibited a significant increase in SDNN following the cognitive task compared to their baseline, an effect not observed in the control group. We observed no significant change for any of the other conditions. This suggests that, under the present testing conditions, concussed athletes may exhibit an altered autonomic response specific to cognitive stress. Interestingly, the changes in HRV observed during the cognitive task performed in isolation disappeared when the same task is combined with a motor task, regardless of whether it is preceded by aerobic exercise. This finding raises questions about the interplay between cognitive and physical demands and how they influence autonomic regulation in concussed individuals.

To our knowledge, this is the first study to employ a dual-task protocol combining both motor and cognitive demands to investigate autonomic response in concussed athletes in the chronic phase of their injury as well as measuring the influence that aerobic effort may have on the autonomic response following this same dual task.

Previous research indicates that concussed athletes exhibit persisting alterations in HRV under conditions of physical stress. For instance, Abaji et al. [24] found that, during an anaerobic handgrip task, concussed athletes showed a significant increase in the LF/HF ratio and a decrease in HF power compared to healthy participants. This was attributed to a decrease in parasympathetic activity. Another study found that concussed athletes require significantly more time for their SDNN values to return to baseline compared to control participants following 20 min of aerobic exertion on a cyclometer, highlighting autonomic alterations [21]. Studies conducted in the acute phase corroborate these results, also showing a reduction in HRV during a physical task [37,38].

In contrast, studies examining autonomic responses to cognitive effort report an increase in HRV among concussed athletes. Harrison et al. [23] found that chronically concussed athletes displayed a significant increase in SDNN and RMSSD compared to healthy peers when performing a switch task, which the authors attributed to increased parasympathetic activity during cognitive engagement. Similarly, concussed athletes in the acute phase of their injury exhibited lower HF power at rest, which increased during a 2-Back cognitive task to levels comparable with healthy athletes [44]. These findings are consistent with our results, showing that, when the switch task is performed on its own, athletes display a significant increase in SDNN.

The neuroanatomical mechanisms underlying long-term autonomic dysfunctions following a concussion have yet to be fully understood. However, evidence suggests that concussion may disrupt the integrity of structures within the central autonomic network (CAN), which may lead to persistent autonomic dysfunctions [34,36,40,72]. Considering the distinct autonomic responses to cognitive and motor tasks observed in concussed athletes, some authors have theorized that alterations in pathways connecting prefrontal regions (responsible for high-level cardiovascular integration) and the CAN could be particularly vulnerable to concussion [32,37,73,74]. The prefrontal cortex plays a critical role in modulating autonomic output through its extensive connections with limbic structures and the brainstem, exerting inhibitory control over sympathetic activity [30,36]. The disruption of these connections may impair this inhibitory control, leading to imbalances characterized by sympathetic hyperactivity and reduced parasympathetic activity.

Huang et al. [44] proposed that, in the acute phase of concussion, a reduction in prefrontal inhibitory control over sympathetic activity could be the underlying mechanism responsible for lowering HRV [30,75]. This would increase sympathetic activity, while reducing parasympathetic activity. This hypothesis finds some support in neuroimaging studies. For instance, using fMRI, Johnson et al. [34] found decreased functional connectivity in the frontal lobe of concussed individuals, suggesting impaired neural communication that might affect autonomic regulation. Additionally, using fNIRS, Kontos et al. [76] observed decreased activation in frontal regions during cognitive tasks in concussed athletes, suggesting impaired cortical processing that could also influence autonomic control.

This reduction in prefrontal engagement could also explain the limited ability of the autonomic nervous system to adapt following physical stress. However, it does not account for the inverse phenomenon observed in autonomic responses to cognitive stress. To address this, Huang et al. [44] proposed that the increase in HRV following a cognitive task may be attributed to a significant rise in prefrontal activity induced by the cognitive demand. This heightened prefrontal activation likely restores cortical inhibitory control over the sympathetic nervous system, thereby shifting the balance toward greater parasympathetic dominance.

This mechanism provides a plausible explanation for the present findings, wherein the rise in HRV observed after the cognitive task dissipates and returns to baseline levels following the dual task, which combines both cognitive and physical components. The integration of cognitive and motor challenges in a dual-task context may effectively neutralize their independent effects on HRV, as each task exerts opposing influences. This interaction would account for the absence of significant differences in HRV observed under dual-task conditions.

Nevertheless, this hypothesis remains speculative and warrants further investigation in future studies. Research should aim to elucidate the precise neurophysiological mechanisms underlying dual-task interactions, as well as their implications for autonomic regulation. Longitudinal and experimental designs integrating neuroimaging and physiological monitoring could provide valuable insights into how cognitive and physical stressors dynamically interact to influence autonomic nervous system functioning.

Given that we observed a moderate effect size, this warrants further exploration of this condition in future studies. Investigating it further could eventually lead to the development of a relevant clinical tool.

Our study is not without limitations, which should be carefully considered when interpreting the findings. First, the absence of an HRV assessment following a “walking-only” condition limits our ability to isolate the specific contribution of the motor task in the dual-task context. Future research employing dual-task protocols should incorporate this condition to disentangle the independent effects of motor and cognitive components. Second, we were unable to fully control for circadian influences on HRV due to participants’ academic schedules, which introduced potential variability into our measurements. Circadian rhythms are known to impact autonomic activity, and standardizing testing times would enhance the reliability of future studies [77]. Third, the sport practiced might have influenced the impact of concussion on the autonomic response during the task. Also, some could argue that, because our participants come from various sports backgrounds and different institutions, this could have an impact on our findings.

Fourth, although we aimed to include female participants, a combination of equipment malfunctions during testing and exclusion criteria (e.g., heart rate being higher during the dual task than during the 20 min aerobic exertion) led to the exclusion of many female participants. This reduced the gender diversity of our sample and limits the generalizability of our findings. A larger sample size would allow future studies to stratify participants by gender, providing a better understanding of potential gender-related differences in autonomic responses following concussion. Finally, our sample consisted exclusively of high-level athletes, whose fitness levels are known to influence HRV [78]. As a result, caution is needed when extrapolating these results to the general population.

## 5. Conclusions

Our study adds to the limited research investigating the long-term consequences of concussion on the ANS. By employing a dual-task protocol that combines motor and cognitive demands, we explored autonomic responses in chronically concussed athletes under conditions somewhat more similar to what athletes experience on the field. Our findings indicate that the switch task is effective in uncovering ANS alterations in concussed athletes, which were not apparent at rest. However, when paired with a motor task, this effect seems to dissipate. Future studies should consider these findings when designing protocols to assess HRV and ANS function in concussed individuals. Our study further contributes to the existing literature demonstrating the relevance of a cognitive task to uncover ANS dysfunctions in chronically concussed athletes, as those deficits may not emerge under resting conditions. HRV is a promising biomarker, which, if utilized under the appropriate conditions, may eventually provide clinicians with objective information on the readiness of student athletes in their return to play and return to learn.

## Figures and Tables

**Figure 1 jfmk-10-00115-f001:**
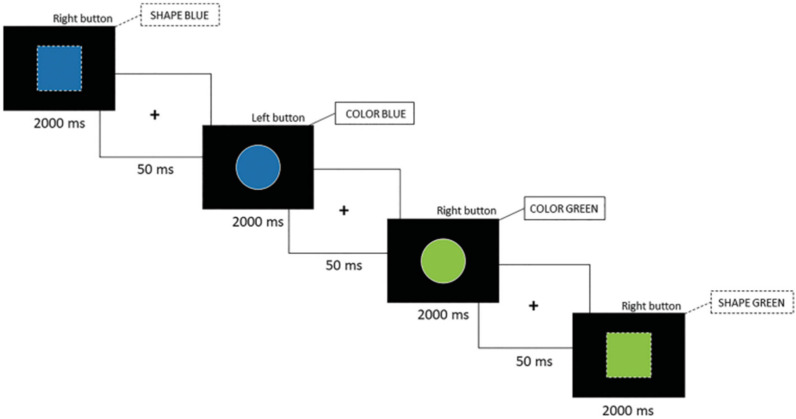
Color–shape switch task (ms = milliseconds).

**Figure 2 jfmk-10-00115-f002:**
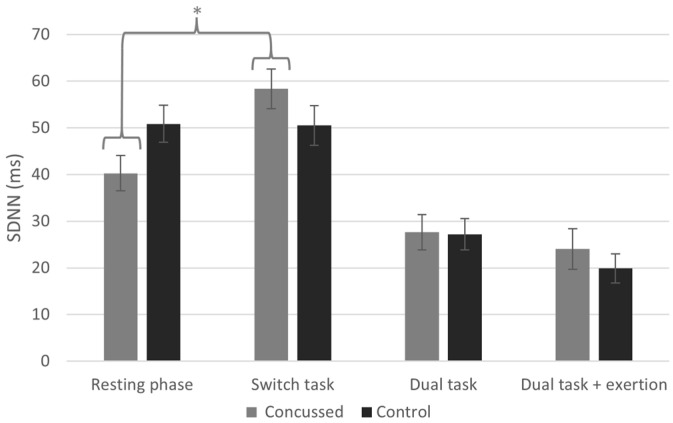
Standard deviation of NN intervals (SDNN) across conditions (* indicates statistically significant difference between conditions, *p* < 0.05).

**Table 1 jfmk-10-00115-t001:** Demographic information (mean ± SD).

	Concussion (*n* = 16)	Control (*n* = 18)
Age (years)	20.90 ± 2.55	19.41 ± 1.62
Height (m)	1.79 ± 0.049	1.83 ± 0.080
BMI (kg/m^2^)	26.36 ± 3.19	24.44 ± 3.35
Time since concussion (months)	5.46 ± 2.00	NA
Education level	9 Universities	4 Universities
	7 Colleges	14 Colleges
Sport	9 Rugby	2 Rugby
	2 Football	3 Football
	1 Ultimate frisbee	3 Volleyball
	1 Hockey	1 Hockey
	1 Soccer	4 Soccer
	1 Basketball	5 Basketball
	1 Cheerleading	

Note: (NA, not applicable; SD, standard deviation; BMI, body mass index; kg, kilograms; m, meters).

## Data Availability

The data presented in this study are available on request from the corresponding author due to ethical reasons.

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
