# Peer review of "Investigating Persistent Sympathovagal Dysregulation Following a Complex Dual Task in Concussed Athletes"

_jfmk, 2025, doi:10.3390/jfmk10020115_

Round 1
Reviewer 1 Report
Comments and Suggestions for Authors
This manuscript provides solid findings on autonomic dysfunction in concussed athletes, offering valuable insights for rehabilitation and return-to-play decisions. The study is well-structured and relevant, but the results could be explored further—especially by analyzing symptom patterns in more detail. A deeper look at PCSS symptoms and their link to autonomic responses could add even more depth to the findings and help better understand post-concussion effects.

Reviewer 2 Report
Comments and Suggestions for Authors
Thank you for submitting our manuscript titled "Investigating Persistent Autonomic Dysfunction After a Complex Dual Task in Concussed Athletes" to the Journal of Functional Morphology and Kinesiology (JFMK). After a thorough evaluation, I provide detailed feedback, structured as follows.
(a) Relevance of the Study: Your study addresses an important and timely issue regarding autonomic dysfunction following sports-related concussions. The implications for athlete health, performance, and clinical management of concussion recovery are significant, particularly concerning return-to-play decisions and the monitoring of long-term athlete health.
(b) Originality: This manuscript demonstrates originality through the innovative integration of cognitive and motor tasks within a dual-task paradigm, coupled with the measurement of autonomic responses using HRV. However, this study could be enhanced by including a purely motor condition as a comparative baseline, clarifying autonomic responses specifically related to cognitive versus motor demands.
(c) Main Results: The key finding that concussed athletes exhibited a significant increase in SDNN following the cognitive task compared to baseline, an effect not observed under dual-task conditions, is compelling. This result highlights the unique autonomic patterns specific to cognitive load and offers important insights into concussion recovery processes.
(d) Comments on Materials and Methods: Your methodology is robust, clearly articulated, and employs gold-standard tools for HRV measurement (Polar H10, Kubios Software). The suggested improvements include the following.
- This provides explicit justification for the selection of specific HRV metrics.
- Addressing potential circadian rhythm influences explicitly.
- Clarifying the rationale behind the selected aerobic exercise intensity.
- The inclusion of female athletes should be considered for broader generalizability.
- Discuss why a purely motor task condition was not included and how future research might address this gap.
(e) Conclusion: Overall, the manuscript presents valuable insights into autonomic dysfunction following concussion with strong potential clinical relevance. Addressing the suggested methodological enhancements and minor improvements in clarity will further strengthen the manuscript.
Major comments:
Title: The title clearly conveys the scope and purpose of the study; however, the specific nature of autonomic dysfunction was measured (e.g., parasympathetic/sympathetic dominance) for greater precision.
Abstract: Your abstract is well structured, succinct, and informative. It effectively outlines the objectives, methods, key results, and conclusions of this study. Nonetheless, you can briefly mention the potential clinical relevance of your findings in the concluding sentence to enhance its impact.
Introduction: The introduction is adequately comprehensive and clearly presents background information, rationale, and relevance. The literature review is thorough, highlighting the gaps that this study aims to address. However, consider briefly summarizing the controversies or conflicting evidence in the existing literature regarding autonomic dysfunction in concussion to further justify your research.
Materials and Methods: Your methods are robust, detailed, and clear. You employ recognized gold-standard tools (Polar H10, Kubios software) suitable for HRV measurements. The inclusion and exclusion criteria were rigorous and controlled for significant confounders. However, some aspects can be improved.
- Clarify the rationale behind the selection of specific HRV metrics (SDNN, RMSSD, LF/HF).
- Explicitly discusses measures or controls for circadian rhythm influence.
- Justify the selected aerobic exercise intensity (80-90% theoretical maximum HR).
- This clearly explains why a purely motor condition was not included as the control.
Statistical Analysis: Statistical procedures (ANOVA, Bonferroni correction) were appropriately chosen and executed correctly. The analysis adequately addressed these assumptions and potential violations. Nevertheless, we explicitly discuss effect sizes in relation to clinical relevance in the Results and Discussion sections.
Results: The results are clearly presented, thoroughly detailed, and logically organized. The narrative effectively complements the presentation of visual data. However, we should emphasize the clinical significance of the statistical findings (particularly the observed SDNN increase).
Tables and Figures: Tables and figures are appropriately designed, clear, and easy to interpret. All units and abbreviations have been fully explained in the figure legends or footnotes to improve clarity.
Discussion: The discussion effectively interprets the findings, relating them comprehensively to previous literature. Your explanations of the mechanisms underlying the observed results are insightful. However, expand on how these findings specifically advance the understanding of concussion recovery and return-to-play assessment.
Limitations: You clearly acknowledge the key limitations (absence of motor-only condition, circadian rhythm, lack of female participants, and high fitness level of the sample). Consider explicitly stating how future studies may address these issues in order to strengthen their implications.
English Language: The manuscript is generally well written, clear, and professional. Minor language adjustments may improve readability as follows:
- Consistently using the active voice for clarity (e.g., “We measured…” rather than “Measurements were taken…”).
- Clarify complex sentence structures to enhance readability.
- Ensure consistent use of terminology throughout (e.g., “autonomic dysfunction” vs. “ANS impairment”).
Reviewer 3 Report
Comments and Suggestions for Authors
Summary:It is imperative that the acronyms are removed and their meanings are included. The methodology and data analysis should be outlined briefly.
Introduction
Lines 46-49 should cite studies that support claims made in the study, in order to enhance its relevance and scientific value.
Likewise, the citation of studies in line 59 is necessary to support claims and enhance the study's relevance and scientific value.
Finally, on line 83, the citation of further studies should once again be included, in order to provide the study with greater relevance and scientific value.
The same is true for lines 109-110, where studies that support the claims should be cited, thus enhancing the relevance and scientific value of the study. It is important to note that the demonstration of this foundation is not evident in the study's conclusion.
The same applies to lines 111-112, which should cite studies that support the claims made in the study.
Furthermore, lines 132-136 should cite studies that support their claims, thereby enhancing the relevance and scientific value of their study. The authors raise issues that appear to lack a scientific basis.
Furthermore, lines 137-140 should cite studies that support their claims, thereby enhancing the relevance and scientific value of their study. The statements made in these lines "seem" to lack a scientific basis, and the discussion of protocols is lacking in clarity, as the specific protocols to which they refer are not indicated.
Materials and Methods
The methodology employed in the selection of the sample must be elucidated, with particular reference to the rationale behind the selection process, ensuring that the subjects are not of an equivalent sporting and educational background, thus facilitating a meaningful comparison. This aspect must be elucidated with great precision, as it has the potential to compromise the integrity of the sample design.
The rationale behind the inclusion of university and college students as a distinct elite category must be thoroughly articulated.
Lines 184: The text should provide a rationale for the selection of coaches as opposed to researchers to make the selection, and also explain how the coaches can ascertain that the players are suffering from concussion. This evaluation should be conducted by a medical professional, as it is not within the purview of a coach, who lacks the necessary expertise in this domain.
In addition, the authors should cite the sources of the questionnaires used in the study, as it appears that they are employed due to their scientific validity, yet no references are provided to support this assertion.
In addition, the citation of sources is imperative for the credibility of research, and this should be addressed in the lines 210. It would be beneficial to elucidate the meaning of a broad questionnaire. , validity, creation... because the inclusion criteria are determined beforehand and have nothing to do with questions but with validated tests.
Results:
The authors employ a comparison of their sample based on the type of sport, which may have a greater relationship with concussion problems compared to others. However, they do not analyse this or the educational level. The reason for this should be explained.
Discussion
Lines 421-430 and 431-437: the presentation of data is of great interest, however, the relevance of the data to the results is not immediately apparent. It is my conviction that the data does not measure the acute phase. If this is the case, then the relevance of the authors' assertions to the results is called into question. This relationship, if it exists, should be elucidated and clarified, or alternatively, if it is determined to be superfluous, it should be eliminated from the analysis.
Furthermore, the lines 439-451 fail to provide a discussion, as they do not relate the findings to other research. In order to support their claims, the authors should refer to scientific literature.
Conclusions
Correct
